# Unraveling Physical and Chemical Effects of Textile Microfibers

**Samantha N. Athey** [1,2], **Bethanie Carney Almroth** [3], **Elise F. Granek** [4], **Paul Hurst** [4], **Alexandra G. Tissot** [4] and **Judith S. Weis** [5,*]

1. Plastic Ocean Project, Wilmington, NC 28403, USA
2. Department of Environmental Sciences, University of North Carolina Wilmington, Wilmington, NC 28403, USA
3. Department of Biological and Environmental Sciences, University of Gothenburg, 405 30 Göteborg, Sweden
4. Department of Environmental Science and Management, Portland State University, Portland, OR 97201, USA
5. Department of Biological Sciences, Rutgers University, Newark, NJ 07102, USA
* Correspondence: jweis@newark.rutgers.edu

**Abstract:** Microfibers are the most prevalent microplastics in most terrestrial, freshwater, and marine biota as well as in human tissues and have been collected from environmental compartments across most ecosystems and species sampled worldwide. These materials, made of diverse compound types, range from semi-synthetic and treated natural fibers to synthetic microfibers. Microfibers expose organisms across diverse taxa to an array of chemicals, both from the manufacturing process and from environmental adsorption, with effects on organisms at subcellular to population levels. Untangling the physical versus chemical effects of these compounds on organisms is challenging and requires further investigations that tease apart these mechanisms. Understanding how physical and chemical exposures affect organisms is essential to improving strategies to minimize harm.

**Keywords:** aquatic; marine; microplastic; semi-synthetic; synthetic; terrestrial; virgin fibers; weathered fibers

## 1. Introduction

Microplastics have been documented in nearly every environment on Earth, including deep-sea trenches, freshwater lakes and rivers, groundwater, as well as the atmosphere *inter alia* [1]. The majority of microplastic particles reported in marine, freshwater, and terrestrial biota are microfibers. While there is currently no standard definition for 'microfiber', the definition currently proposed by the US National Oceanic and Atmospheric Administration (NOAA) is as follows: microfibers are polymeric fibrous particles (<5 mm) that have been chemically modified and have a length to width aspect ratio of 3:1 [2]. Microfibers in the environment can be composed of a variety of materials. Synthetic fibers account for nearly 14% of global plastic production [3], and approximately 60% of textiles are produced using synthetic materials, such as polyester, nylon, polyamide, etc. [4,5]. These materials, similar to other microplastics, are derived from fossil fuels and sometimes feedstocks consisting of recycled content.

While a large proportion of microfibers reported in environmental samples are composed of plastic or synthetic materials, an equally large (sometimes larger) proportion of anthropogenic microfibers found are composed of semi-synthetic (i.e., rayon) and natural materials (i.e., wool, cotton) [1]. Semi-synthetic fibers, while derived from natural materials, are chemically processed and formed via extrusion, similar to synthetic fibers. 'Natural' textile fibers are also derived from natural materials, and, while they do not undergo the same extrusion process as semi-synthetic fibers, they can contain a suite of chemical additives—dyes and finishing agents incorporated during production and manufacturing (discussed further below) [5–7].

Sources of these fibers to the environment can vary. It is currently hypothesized that a majority of microfibers are shed from textiles (i.e., clothing, upholstery, carpeting) during

production and manufacturing [7,8], consumer use (i.e., laundering and wear [1,9,10], and following disposal [3,11]. Other sources of microfibers include derelict fishing gear, tire wear, cigarette filters, geotextiles, and personal care products (e.g., wet wipes, face masks) [12–15].

While most microplastics documented in environmental samples are microfibers, the majority of experimental studies related to the effects of microplastics expose organisms to microspheres (or beads), pellets, or fragments, which can be easily purchased at specified sizes and polymer types [16]. Far fewer studies have used microfibers [16–19]. When the effects of microfibers are compared to those of non-fibrous particles (i.e., spheres, fragments, pellets), fibers tend to be more toxic [16]. Further, most studies on microfibers have focused on the effects of synthetic fibers, whereas the impacts of natural and semi-synthetic fibers are understudied. Yet, when investigated, natural and semi-synthetic fibers have comparable effects to those of their synthetic counterparts [20,21]. Furthermore, many experimental studies on fibers use exposure concentrations considerably higher than those found in the environment and tend to expose organisms for short periods of time. Further, given the discrepancy between the conditions used in experimental studies and conditions that occur in the environment, it is difficult to make conclusions about the actual effects of microfibers (Figure 1) [22].

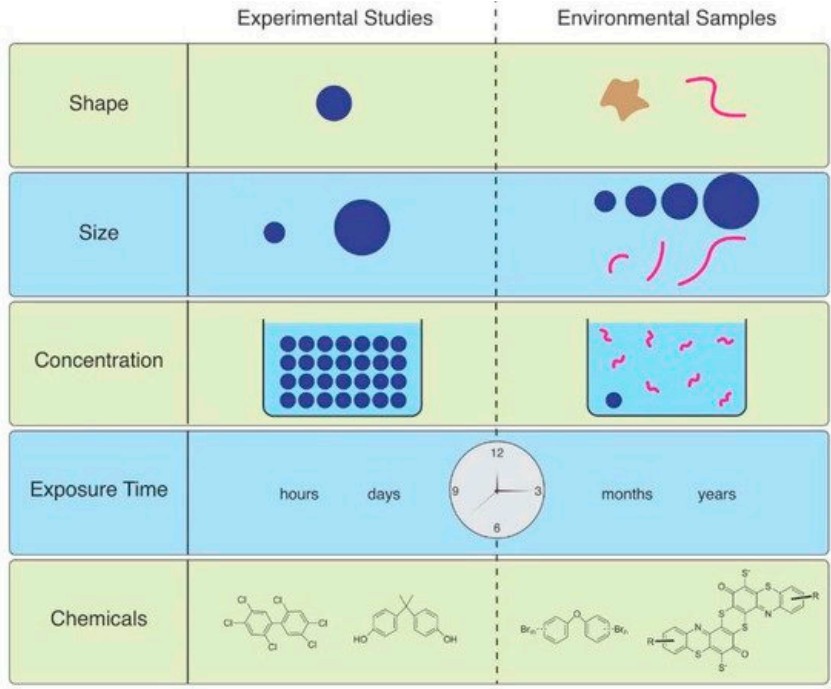

**Figure 1.** Differences between environmental conditions and experimental microplastic studies. From Weis and Palmquist, 2021. Open access.

Microfibers can vary in the polymeric materials of which they are composed, as well as the suite of chemicals that are intentionally added during production (i.e., chemical additives, dyes, and finishes) and unintentionally accumulated from the environment (i.e., persistent environmental contaminants) [23–26]. Many of these chemicals are known to be carcinogenic, mutagenic, and/or endocrine-disrupting compounds (EDCs) and can potentially leach from fibers to the surrounding environment [12,27]. Once in the environment, persistent environmental contaminants, such as heavy metals, PCBs, PAHs, can adsorb to fibers, causing "weathered" (or environmentally exposed) fibers to have different associated chemical profiles and therefore different toxicity than "virgin" fibers [28]. Given their high surface area to volume ratio and demonstrated high sorptive capacity for contaminants [24], microfibers may be a vector for chemical exposure in biota.

In this paper, we summarize the effects of microfibers in an attempt to unravel the impacts due to their physical nature versus those caused by their capacity to act as vectors for chemical exposure.

## 2. Effects of Microfibers

### 2.1. Synthetic Fibers

Early observations from occupational exposure in industries, including textile production and plastic manufacturing [29–33], indicated that exposure occurred in humans and caused inflammation of the lungs; moreover, surgical exposure to microplastics can decrease a patient's immune response [32] (Melgert et al., 2021). Recent studies document microplastic fibers in human intestinal tracts, placental tissue, blood, and lung tissue, though effects on humans remain largely unknown [34–37]. Studies documenting microplastic ingestion from food items and drinking water [38–41] and inhalation both indoors and outdoors [42,43] demonstrate the ubiquitous nature of these contaminants.

The reported effects of synthetic fibers on aquatic and terrestrial organisms are growing, and our current knowledge indicates impacts that range from subcellular to community levels [44] (Figure 2, Table 1). Effects have been documented in aquatic taxa including fish, Crustacea, Mollusca, Echinodermata, and Rotifera [45], as well as terrestrial organisms, including insects, Annelid worms, and Nematodes [46–48] (Table 1). Exposure to synthetic fibers can affect subcellular and cellular level processes, including altered gene expression and enzyme activity, DNA damage, and the retention of zinc [47]. For example, both juvenile and adult sea cucumber, *Apostichopus japonicus*, experience altered acid phosphatase and alkaline phosphatase activity levels—key biomarkers of immune health—and oxidative stress after exposure to environmentally relevant concentrations of synthetic microfibers [49]. Once in the bloodstream, microfibers can translocate to other organs [50], affecting tissue and organ systems, but accumulation in the gut itself can cause effects. For example, nylon microfibers (10–100 μm) can accumulate in the gut of the shore mussel *Mytilus edulis* and affect the long-term clearance rate of phytoplankton biomass from the water column, resulting in a 21.3% decrease in phytoplankton removal ability after exposure to microfibers [51]. At the organismal level, synthetic fibers can cause physical and neurological damage across an array of terrestrial, aquatic, and marine species. Environmentally relevant concentrations (0.1 mg/L) of 500–10,000 μm polystyrene microfibers resulted in the decreased photosynthetic activity of symbionts in Acroporid corals, triggering coral stress responses [52,53]. Though research on the biological consequences of microfiber exposure is sparse, the diverse array of effects on subcellular to organ level processes may be predictive of population- and community-level effects.

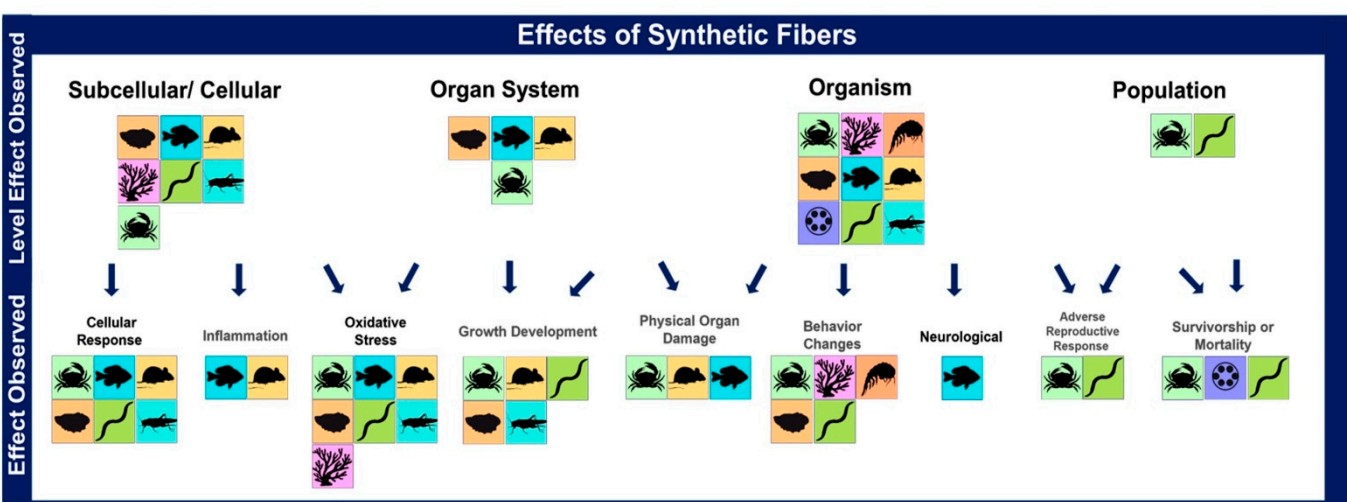

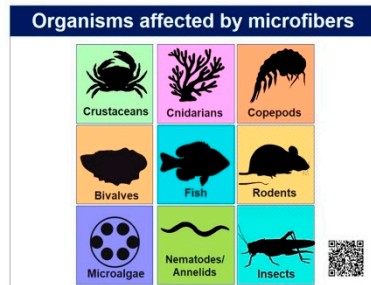

**Figure 2.** The effects of synthetic fibers on organism health have been documented across biological levels in a diverse array of species. Slanted arrows show an effect on more than one level. From Granek et al., 2022.

However, reducing the effects of fibers in the environment is not as simple as switching from synthetic to more natural alternatives. Evidence suggests bioplastics and natural textile materials produce effects similar to synthetic plastics when degrading in the environment [20,54,55].

**Table 1.** Ecological and biological effects of fibers on organisms by species and material type. Modified from Granek et al., 2022.

| Level | Type of Effect | Organism Clade | Genus/Species | Plastic Material Type | In Text Citation |
|---|---|---|---|---|---|
| Sub/Cellular | Adverse Immune Response | Annelid Worms, Bivalves, Corals | *Eisenia andrei* | Polyamide nylon, polyproplene | [52,56] (Cole et al. 2020); |
| | | | *Mytilus* spp. | | |
| | | | *Acropora* sp. | | |
| | | | *Seriatopora hystrix* | | |
| | Cellular Response | Annelid Worms, Bivalves, Coral, Crustaceans, Humans, Nematodes, Rodents | *Lumbricus terrestris* | Composite household lint, nylon, polyester, polypropylene, polyethylene terephthalate (PET) | [48,52,56–62] |
| | | | *Eisenia andrei* | | |
| | | | *Mytilus galloprovincialis* | | |
| | | | *Acropora* sp. | | |
| | | | *Seriatopora hystrix* | | |
| | | | *Nephrops norvegicus* | | |
| | | | *Homo sapiens* | | |
| | | | *Caenorhabditis elegans* | | |
| | | | *Mus musculus* | | |
| | Oxidative Stress | Annelid Worms, Bivalves, Coral, Echinoderms, Fish, Nematodes, Terrestrial Snails | *Lumbricus terrestris* | Polyester, polyamide nylon, polyethylene (80%), polyester (19%), rayon (1%), polypropylene, polyethylene terephthalate (PET) | [48–50,52,57,63,64] (Cole et al., 2020); |
| | | | *Mytilus* spp. | | |
| | | | *Acropora* sp. | | |
| | | | *Seriatopora hystrix* | | |
| | | | *Apostichopus japonicus* | | |
| | | | *Dicentrachus labrax* | | |
| | | | *Trachurus trachurus* | | |
| | | | *Scomber colias* | | |
| | | | *Danio rerio* | | |
| | | | *Caenorhabditis elegans* | | |
| | | | *Achatina fulica* | | |
| Organ | Growth Development | Bivalves, Crustaceans, Fish | *Mytilus galloprovincialis* | Composite household lint; ethylene vinyl acetate (EVA); polypropylene, polyethylene terephthalate | [20,60,65,66] |
| | | | *Emerita analoga* | | |
| | | | *Artemia franciscana* | | |
| | | | *Carassius auratus* | | |
| | Inflammation | Fish; Rodents; Zooplankton | *Carassius auratus* | Ethylene vinyl acetate (EVA), polypropylene, polyester, polyethylene terephthalate (PET) | [29,63,65,67] |
| | | | *Danio rerio* | | |
| | | | *Cavia porcellus* | | |
| | | | *Artemia franciscana* | | |
| | Oxidative Stress | Crustaceans, Fish | *Homarus americanus* | Polyethylene terephthalate (PET), polypropylene | [63,68] |
| | | | *Danio rerio* | | |
| | Physical Organ Damage | Annelid Worms, Bivalves, Crustaceans, Fish, Humans, Rodents, Terrestrial Snails, Zooplankton | *Eisenia andrei* | Composite household lint; polypropylene, polyethylene terephthalate (PET), polyethylene (80%); polyester (19%); rayon (1%) ethylene vinyl acetate (EVA), polycarbonate, polyamide, polyester | [20,29,34,50,56,58,60,62–65,67,69] |
| | | | *Mytilus galloprovincialis* | | |
| | | | *Artemia franciscana* | | |
| | | | *Nephrops norvegicus* | | |
| | | | *Dicentrachus labrax* | | |
| | | | *Trachurus trachurus* | | |
| | | | *Oryzias latipes* | | |
| | | | *Scomber colias* | | |
| | | | *Carassius auratus* | | |
| | | | *Danio rerio* | | |
| | | | *Homo sapiens* | | |
| | | | *Cavia porcellus* | | |
| | | | *Achatina fulica* | | |
| | | | *Artemia franciscana* | | |

**Table 1.** *Cont.*

| Level | Type of Effect | Organism Clade | Genus/Species | Plastic Material Type | In Text Citation |
|---|---|---|---|---|---|
| Organism | Adverse Reproductive Response | Bivalves, Crustaceans, Nematodes, Terrestrial Vegetation, Worm, Zooplankton | *Mytilus galloprovincialis* / *Daphnia magna* / *Emerita analoga* / *Caenorhabditis elegans* / *Lolium perenne* / *Aporrectodea rosea* / *Ceriodaphnia dubia* | Polypropylene, polyethylene terephthalate (PET), high-density polyethylene (HDPE), polylactic acid (PLA), polyacrylicnitrile (PAN) | [48,62,70–74] |
| | Behavioral Change | Annelid Worms, Bivalves, Cnidarians, Crustaceans, Nematodes, Terrestrial Snails, Zooplankton | *Lumbricus terrestris* / *Mytilus galloprovincialis* / *Mytilus edulis* / *Macomona liliana* / *Aiptasia pallida* / *Hyalella azteca* / *Calanus finmarchicus* / *Gammarus fossarum* / *Nephrops norvegicus* / *Caenorhabditis elegans* / *Achatina fulica* / *Daphnia magna* / *Tigriopus japonicus* | Polyester, composite household lint, nylon, polyethylene terephthalate (PET), polypropylene, polyamide (PA) | [38,48,51,57,58,60,64,66,68,72,75–78] (Kang et al., 2020); (Lahive et al., 2022) |
| | Growth Development | Bivalves, Crustaceans, Microphytobenthos, Nematodes, Terrestrial Veg., Worm, Zooplankton | *Macomona liliana* / *Hyalella azteca* / *Emerita analoga* / *Carcinus maenas* / *Calanus finmarchicus* / *Nephrops norvegicus* / *Homarus americanus* / *Cyanobacteria* / *Caenorhabditis elegans* / *Allium fistulosum* / *Lolium perenne* / *Aporrectodea rosea* / *Lactuca sativa* / *Daucus carota* / *Daphnia magna* / *Artemia franciscana* / *Ceriodaphnia dubia* | Polyethylene terephthalate (PET), polypropylene, high-density polyethylene (HDPE), polylactic acid (PLA) nylon, polyester | [38,48,58,66–68,70–72,77–82] |
| | Neurological | Fish | *Dicentrachus labrax* / *Trachurus trachurus* / *Scomber colias* | Polyethylene (80%), polyester (19%), rayon (1%) | [50,83] |
| | Survivorship or Mortality | Annelid Worms, Crustaceans, Zooplankton | *Eisenia andrei* / *Hyalella azteca* / *Emerita analoga* / *Palaemonetes pugio* / *Artemia franciscana* / *Nephrops norvegicus* / *Homarus americanus* / *Daphnia magna* / *Ceriodaphnia dubia* | Polypropylene, polyethylene terephthalate (PET), polyester, lyocell | [20,56,58,68,71–73,77,84,85] |
| Population | Adverse Reproductive Response | Crustaceans, Nematodes | *Emerita analoga* / *Caenorhabditis elegans* | Polypropylene, polyethylene terephthalate | [48,72] |

## 2.2. Semi-Synthetic and Natural Fibers

Despite being derived from natural materials, many studies report the widespread presence of semi-synthetic and natural textile fibers in marine, freshwater, and terrestrial biota [1] (*inter alia*). When reported, these fibers often constitute the majority of anthropogenic microfibers present in a sample [86,87]. Once in the environment, natural fibers can be biodegraded by factors such as naturally occurring microbes that consume cellulose,

aerobic degradation, or enzymatic breakdown in soils [6,88,89]. Although natural and semi-synthetic fibers degrade faster than their synthetic counterparts in the environment [90], these fibers are sufficiently persistent to undergo long-range transport and accumulate in sensitive ecosystems [1,87]. Additionally, the chemicals incorporated into non-plastic fibers may prolong their persistence in the environment [12,91]. The decreased environmental persistence of non-plastic fibers when compared to their synthetic counterparts does not necessarily translate into reduced toxicity, given the potential for the physical impacts of the fiber upon exposure [6,92–94]. Further, chemical treatments intentionally applied during the production and manufacturing of cotton and wool textiles, such as those discussed below, combined with a higher adsorption capacity for unintentionally accumulated chemicals, could potentially mean an increased toxicity for natural fibers compared to synthetic fibers [23–25,94–96]. To our knowledge, only three studies have compared the toxicological effects of microfibers composed of natural and semi-synthetic to synthetic materials [19–21], though others are ongoing. These investigations suggest that the organismal effects of microfibers created from natural polymers (e.g., cotton) are equivalent or only slightly reduced compared to synthetic fibers. However, more testing involving a variety of natural, semi-synthetic, and synthetic polymers is needed to better understand the role of polymer composition in toxicity. Given the physical properties of natural fibers and their frequent detection in the environment, they may pose a risk to organisms in natural systems and warrant further study.

## 3. Microfibers as Vectors for Chemical Exposure

While known effects of microfibers are described above, these studies do not delineate physical effects of the fibers themselves from potential impacts of associated chemical contaminants [97]. Microfibers, including semi-synthetic and natural materials, present a complex mix of physical and chemical properties that can influence toxicity. These include the material (i.e., polymeric) composition of the particle, the size, shape, density, and surface properties of the particle, as well as the profile of associated chemical contaminants [98,99].

The textile industry is regarded as one of the most chemical-intensive industries on the planet. Thousands of chemicals are registered for use in the production, assembly, storage, and shipping of textile fibers, including dyes and finishing products [23,100,101]. Not only are chemicals intentionally incorporated into textile fibers during production and use, chemical contaminants (e.g., PCBs, PBDEs) may also be unintentionally sorbed from the surrounding environment [24,102,103]. Here, we explored a major outstanding question pertaining to the ecological and human health impacts of microfibers: the capacity of microfibers to act as vectors for toxic chemical exposure.

### 3.1. Chemical Usage and Accumulation on Textile Fibers

Chemicals are used throughout the textile production process, from the harvesting of raw materials to finishing and storage. The application of synthetic materials used in the production of plastic textile fibers begins with the extraction of fossil fuels and the manufacturing of plastic monomers. Polymer products, used to create synthetic textile fibers, are created from monomers via polymerization reactions. Following their creation, it is possible for unreacted monomers and intentionally added substances (e.g., titanium (III) chloride, antimony), which drive polymerization reactions, to remain on finished polymer products. Additionally, non-intentionally added substances, including reaction byproducts, degradation products, and contamination, may be incorporated into synthetic polymer products. Despite being derived from natural sources, semi-synthetic and natural textile fibers also undergo heavy chemical processing and thus cannot be considered inherently natural or environmentally friendly [104]. These include substances used in the cultivation of plant and animal fibers (e.g., herbicides, insecticides, rodenticides), as well as the chemical processing used to create regenerated or semi-synthetic fibers [105].

While many chemicals are used in the cultivation and synthesis of textile fibers themselves, the bulk of chemicals used during production are applied to constructed garments.

These include pigments and dyes, wrinkle-resistance finishing, antimicrobial agents, and water and stain repellents [23,106]. Many of the finishing products that are applied to textiles are persistent, bioaccumulative, and toxic (PBT) substances and were identified as chemicals of concern due to their potential impacts on human and environmental health [107]. The largest and most diverse group of chemicals applied to textiles during production are pigments and dyes (i.e., colorants). Human health effects of these chemicals vary and can include allergic reaction, growth and developmental impacts, as well as carcinogenic effects [108]. Further, through contact with skin, colorants such as azo dyes can produce carcinogenic degradation compounds [109,110]. Azo dyes are the most dominant dye class used in textile production, accounting for 60–70% of the global market [111].

Other known toxicants used in the production of textile fibers include per- and poly-fluorinated alkylated substances (PFOS and PFAS), which are applied to textiles and other consumer products (e.g., food packaging, cookware). Phthalates are another group of chemicals commonly applied to textiles, most often used in polyvinyl chloride (PVC) prints and the coatings of decorative images [112]. Phthalates are well known endocrine-disrupting compounds (EDCs) due to their harmful impacts on reproductive health [113]. The relative importance of microfibers as a source and vector of these chemicals in the environment is not yet quantified, but the textile industry is suspected to be a significant source, e.g., via effluent and water discharge [7,8].

Finally, there are also chemical contaminants that may unintentionally accumulate on our garments from the surrounding environment [24,102,103]. The sorption and desorption of chemicals is dictated by their physical-chemical properties, as well as the physical-chemical properties of the textiles [114]. Chemicals, including those not originally intended for use in textiles, such as PBDEs, phthalates, organophosphate esters (OPEs), polycyclic aromatic hydrocarbons (PAHs), and PCBs, have been documented to accumulate on clothing from contact with air, dust, and/or contaminated products. In fact, textiles have a large sorption capacity for semi-volatile organic compounds (SVOCs) [24,102,103]. It is estimated that the amount of clothing worn by an adult (2 m$^2$) can sequester the equivalent of approximately 100 m$^3$ of air per day (Saini et al. 2016). Further, microfibers released from textiles have been shown to adsorb chemicals, including PAHs and PCBs [115–117].

Although we primarily focused on microfibers derived from textiles here, other sources of microfibers exist, but they are not as well characterized. These include carpeting and personal care products [12–15]. Cigarette filters, composed of cellulose acetate, a semi-synthetic fiber [12,118], are a major non-textile source of microfibers. It is estimated that the 4.5 trillion cigarette filters littered annually generate approximate 0.3 million tons of microfibers each year [12,119]. A suite of toxic chemicals is associated with cigarette filters, including heavy metals, PAHs, etc., and cigarette filter leachates are well known to be toxic [120–122]. Further, Belzagui et al. [12] found that negative effects of cigarette leachate can be exacerbated by the physical effects of the fibers.

Bioplastics such as PLA are offered as a "green" alternative to synthetic and semi-synthetic plastics due to their ability to biodegrade under certain industrial conditions [123] and constitute another suite of understudied semi-synthetic materials for which little is known about their environmental fate and effects. Though bioplastics have not been heavily utilized in textile production, interest is growing among smaller producers such as Xtep. Studying the effects of bioplastics is important to addressing emerging synthetic alternatives and their environmental effects.

### 3.2. Microfiber-Mediated Chemical Release and Exposure

While chemicals associated with microfibers can broadly be categorized as those that are (1) intentionally added during production and use and those that (2) unintentionally accumulate from the environment, most studies investigating the chemical sorption-desorption dynamics of synthetic microfibers focused on the latter category of chemicals. These include pharmaceuticals, heavy metals, and organic contaminants (e.g., PAHs, PCBs) [115,124–126]. While most research on the sorption behaviors of environmental con-

taminants to microplastics has focused on microplastic fragments or spheres [127–129], a few studies demonstrated the sorption and release of chemicals from synthetic and natural textile fibers [28,116,117,130,131].

Several factors likely influence the sorption and desorption of chemicals to microfibers, including the physical properties (i.e., crystallinity, surface area, surface condition) and chemical properties (i.e., polymer type, surface charge, hydrophobicity) of the fiber, as well as the physical-chemical properties of the chemical and surrounding environmental media [116,132]. Additionally, the degree of physical weathering of a fiber in the environment may influence its surface morphology and associated chemical profile [28]. Microfibers, including natural, semi-synthetic, and synthetic fibers, degrade in the environment via photo-degradation. Sait et al. [28] demonstrated that the degree of weathering (measured as changes to surface morphology and fragmentation) varied among different types of fiber (i.e., polyester, acrylic, wool). Further, they identified chemical leachates, including monomers, additives, and degradation products, in both pristine and degraded fibers.

### 3.3. Toxicity of Virgin vs. 'Weathered' Particles

To date, empirical evidence demonstrating the potential importance of microfibers as vectors and/or sources of chemicals to biota remains a major understudied question in the microfiber and microplastics research fields (see [133,134]). Chemical profiles vary between virgin and 'weathered' microplastics, including fibers [28,135,136]. While research on the toxicity of virgin versus 'weathered' fibers is limited, this type of exposure study is critical for delineating between the physical effects of the fibers and their associated chemical profile. Nearly all ecotoxicological testing of this type has investigated the toxicity of virgin versus 'weathered' non-fibrous particles, such as pellets and spheres. These studies suggest that organisms respond differently when particles are exposed to the environment, such that 'weathered' particles sometimes cause greater toxicity compared to virgin particles [137–141]. However, sorption-desorption can differ between non-fibrous microplastics and fibers given differences in physio-chemical properties, sorptive capacities, and chemical profiles [128]. Further, other factors that influence the uptake of chemicals from ingested particles, such as gut residence time, can also vary between fibers and other types of microplastics. Another important consideration for future investigations is the bioavailability of microfiber-sorbed contaminants compared to other exposure pathways [136,142,143]. For example, Beckingham and Ghosh [142] demonstrated that the PCB uptake from microplastic spheres into benthic worms was much lower than the uptake from surrounding sediments. Further, Thaysen et al. [143] reported evidence of the bidirectional transfer of PBDEs from ingested microplastics in seabirds, where in some cases highly contaminated tissues may be a source of contaminants to ingested microplastics. In these cases, microplastics were not a significant vector for chemical exposure. Given the diversity of microplastics, microfibers, and their associated chemical contaminant profile, generalization to this entire contaminant class requires further research.

### 4. Discussion

Microfibers from textiles are ubiquitous contaminants found in all niches on the planet. They are persistent and are able to spread over long distances. While the current state of knowledge concerning the impacts of microfibers on the environment is limited, current evidence indicates that these contaminants have the potential to drive toxicity. Furthermore, semi-synthetic and treated natural fibers are similar to synthetic microfibers, regarding both their environmental fate and hazard properties. However, the persistence of plastic in the environment suggests that synthetic microfibers will remain a problem for a longer period of time than natural fibers.

How can scientists distinguish the chemical from the physical effects? The experimental designs of most studies to date do not allow researchers to make that distinction. Suggestions for improvements were discussed previously [97] and include considering the complex nature of microplastics and categorizing effects as particle-driven hazardous

impacts versus chemical toxicity and designing studies to better understand direct versus indirect effects (e.g., food dilution). Choice of species, medium and exposure matrix, duration, and choice of controls can be improved to better our understanding as we accumulate knowledge in this field.

However, studies comparing weathered vs. virgin microplastics help us understand different drivers, and these studies often report more severe effects from weathered microparticles (see Section 3.3 above) due to the environmental contaminants adsorbed during the weathering process. Dosing microplastics with specific chemicals and then exposing different groups of organisms to dosed or undosed microplastics, in addition to unexposed controls, can further our understanding of chemical toxicity effects. Studies should take careful consideration of polymer type (e.g., rubbery versus glassy) and make use of non-polymer control particles, as well as the physiochemical properties of chemical substances (e.g., Log Kow) and exposure route (water, food, trophic transfer), as shown in studies by Ašmonaitė et al. and Bour et al. [129,144]. Le Bihanic et al. [139] exposed marine medaka *Oryzias melastigma* embryos and larvae to microplastics spiked with benzo(a)pyrene (MP-BaP), perfluorooctanesulfonic acid (MP-PFOS), or benzophenone-3 (MP-BP3) for 12 days. The particles agglomerated on the surface of the egg chorion and did not penetrate it or contact the developing embryos. While embryos treated with virgin microplastics showed no toxic effects, those treated with microplastics with PFOS had decreased survival and did not hatch. Larvae exposed to microplastic particles with BaP or with BP3 had reduced growth, developmental anomalies, and abnormal behavior. These investigators found that, compared to equivalent waterborne concentrations, BaP and PFOS spiked on microplastics appeared to be more embryotoxic than when chemicals were in seawater. These studies used microbeads, rather than microfibers, and demonstrated effects from chemical toxicity.

Careful examination of organisms for microplastics in their tissues after exposure can indicate chemical toxicity when ingestion or translocation are not observed. Zhu et al. [145] fed Japanese medaka (*Oryzias latipes*) diets amended with 500, 1000, or 2000 $\mu$g/g 10 $\mu$m fluorescent spherical polystyrene microparticles for 10 weeks during maturation from juveniles to spawning adults. Microscopic examination, histologic sections, and scanning electron microscopy showed no evidence of any translocation to other internal organs. Nevertheless, females showed dose-dependent decreases in egg number, and histological analysis showed changes in the kidney and spleen. Since no microplastics were found in any tissues, they attributed effects to the leaching of chemical additives such as DEHP (di(2-ethylhexyl)phthalate) from the particles.

Comparisons of synthetic versus 'natural' fibers (animal or plant, i.e., wool or cotton) could provide indications of 'particle'-driven effects, as can studies using particles of different shapes. While we are gaining better understandings of the effects of exposure to synthetic microfibers, the potential toxicity of anthropogenically modified natural or semisynthetic microfibers remains understudied [1]. Microfibers are the shape most commonly identified in organisms, and the majority of studies detect effects on fibers, yet many effect studies focus on spheres, which are infrequently detected in the environment [16]. Some efforts have compared the ingestion and egestion of fibers and spheres in two different species (brine shrimp and stickleback) and found differences between species and particle shapes, e.g., Bour et al. [17]. Future studies trying to unravel these effects may need to include a chemical analysis to ascertain which chemicals leached or desorbed from the microplastic and entered the animals. One possible method for textile microfibers would be to obtain them from white vs. dyed (identical) garments and compare effects. Such a study could provide information about the toxicity of dyes used, though not the other additives in the textiles. These steps are needed to unravel physical and chemical effects.

Even as microfibers from textiles are the dominant microplastic type in the environment, now ubiquitous in all niches, textile production is predicted to increase in the future, and thousands of chemicals are used in production and in finished products. Plastics and associated chemicals are now recognized as planetary boundary threats [146], as the massive quantities produced, largely uncontrolled with minimal transparency from the industry,

are threatening our environment, our health, and our ability to thrive. Production outpaces societies' ability to conduct safety related assessments and monitoring. Improved understanding of the impacts of microfibers on the environment will inform risk assessments and support mitigation strategies, together with a deeper understanding of sustainability and increased circular approaches in the industry, allowing us to decelerate environmental degradation and move back within a safe operating space for humanity [147].

**Author Contributions:** Conceptualization, introduction, J.S.W.; physical effects, E.F.G., A.G.T. and P.H.; chemical effects S.N.A. and B.C.A.; discussion, J.S.W. and B.C.A. All authors have read and agreed to the published version of the manuscript.

**Funding:** This research received no external funding.

**Data Availability Statement:** Not applicable.

**Conflicts of Interest:** The authors declare no conflict of interest.

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
