# Peer review of "Unraveling Physical and Chemical Effects of Textile Microfibers"

_water, doi:10.3390/w14233797_

Round 1

Reviewer 1 Report

Table 2 is it is only partially visible on the document!?

    In the manuscript entitled "Unraveling Physical and Chemical Effects of Microfibers" (water-2038939), submitted to Water, the authors reviewed provided studies of the microfiber effects on organisms and particular organism groups across biological levels. They paid attention to microfibers, and chemical substances from microfibers that are released into the environment from textile materials.

The manuscript contains Abstract, Keywords, Introduction, Effects of microfibers, Microfibers as vectors for chemical exposure, and Discussion sections, Author  Contributions, Founding,  and Conflict of Interest  details, list of References, as well as two Figures, and one Table with corresponding captions.

The manuscript is well written and clear. The Introduction section is quite informative, and effects of microfibers and their role as vectors of different toxic chemical are well described. Finally, the microfiber-driven hazard end effects versus microfiber chemical toxicity (distinguishing related physical and chemical effects) are concisely and clearly discussed. Apart from noticed (and I believe technical) error related to Figure 1 I have no any other suggestions.

Thereby, I believe that the manuscript deserves to be published in Waters after appointed technical correction.

Author Response

The only suggestion related to placement of Figure 1. We understand that the MDPI office can fix this. (It was centered when we sent it in initially) 

Reviewer 2 Report

Dear Editor,

In their study, the authors (II) evaluated the studies on the physical and chemical effects of microfibers on living things, (II) They took a look at the ecological and biological effects of microfibers for some living groups, and (III) in particular, they drew attention to the release of chemicals from microfibers originating from the textile sector and their effects on living things. In this review study, the authors have chosen a current problem that concerns all branches of science as a topic. This paper has a potential to be accepted, but some important points have to be clarified or fixed before we can proceed and a positive action can be taken.

I here summarize this points:

1- Figure 2 is not complete in the text (line 133).

2- Fibers, including microfibers, are found in every ecosystem in nature. For example, in beach sand, in the sea and sea ice, in lakes and rivers, in agricultural land, in the atmosphere. Giving examples of environments where microfibers are found will strengthen the article. For example;

Kanhai, L. D. K., Gardfeldt, K., Krumpen, T., Thompson, R. C., & O’Connor, I. (2020). Microplastics in sea ice and seawater beneath ice floes from the Arctic Ocean. Scientific Reports10(1), 1-11.

Kumar, M., Xiong, X., He, M., Tsang, D. C., Gupta, J., Khan, E., ... & Bolan, N. S. (2020). Microplastics as pollutants in agricultural soils. Environmental Pollution265, 114980.

Liu, S., Pan, Y. F., Li, H. X., Lin, L., Hou, R., Yuan, Z., ... & Xu, X. R. (2022). Microplastic pollution in the surface seawater in Zhongsha Atoll, South China Sea. Science of The Total Environment822, 153604.

Wright, S. L., Ulke, J., Font, A., Chan, K. L. A., & Kelly, F. J. (2020). Atmospheric microplastic deposition in an urban environment and an evaluation of transport. Environment International136, 105411.

Yabanlı, M., Yozukmaz, A., Åžener, Ä°., & Ölmez, Ö. T. (2019). Microplastic pollution at the intersection of the Aegean and Mediterranean Seas: A study of the Datça Peninsula (Turkey). Marine Pollution Bulletin145, 47-55.

3- The sequence numbers are written twice in the list of references.

The content of the article in accordance with the aims of the Water

The article is scientifically sufficient.

The literature sufficiently critical, current, and internationally is evaluated

The language of the article is correct and clear.

The discussion part is quite comprehensive in the paper.

Tables and figures are well designed and necessary.

Acceptable after minor revisions.

Author Response

Reviewer said Fig. 2 was off-center. We understand that the production office of MDPI can fix that problem. (It was centered when we sent it in)                        Reviewer suggested additional references on where microplastics are found. We do not think this is necessary since reference (1) reviews the literature on that. We already have over 130 references, which is a large number, focusing on the topic of effects, the subject of this article.                                                                Reviewer points out that reference numbers are printed twice in the reference list. We have cut out one in the track changes version we are submitting.